# Ductility Estimation for Flexural Concrete Beams Longitudinally Reinforced with Hybrid FRP–Steel Bars

**DOI:** 10.3390/polym14051017

**Published:** 2022-03-03

**Authors:** Rendy Thamrin, Zaidir Zaidir, Devitasari Iwanda

**Affiliations:** Department of Civil Engineering, Faculty of Engineering, Andalas University, Padang 25163, Indonesia; zaidir2000@yahoo.com (Z.Z.); devitasariiwanda@gmail.com (D.I.)

**Keywords:** hybrid FRP–steel reinforcement, ductility, hybrid reinforcement ratio, fiber element, neutral axis

## Abstract

An experimental study was conducted to evaluate the ductility of reinforced concrete beams longitudinally reinforced with hybrid Fiber Reinforced Polymer (FRP)–steel bars. The specimens were fourteen reinforced concrete beams with and without hybrid reinforcement. The test variables were bar positions, ratio of longitudinal reinforcement, and type of FRP. The beams were loaded until failure using a four-point bending test. The performance of the tested beams was observed using the load–deflection curve obtained from the test. Numerical analysis using the fiber element model was carried out to examine the growth of neutral axis due to the effects of the test variables. The neutral axis curves were then used to estimate the neutral axis angles and displacement indices. The test results showed that the reinforcement position did not significantly affect the flexural capacity of beams with a higher ratio of hybrid reinforcement, but was quite significant in beams with a lower ratio of hybrid reinforcement. It was observed from the test that the flexural capacity of beams with hybrid reinforcement was 15–45% higher than that of beams with conventional steel bars, depending on bar positions and the ratio of longitudinal reinforcement. The ductility of beams with hybrid reinforcement was significantly increased compared to that of beams with FRP, but decreased as the hybrid reinforcement ratio (*A_f_*/*A_s_*) increased. This study also showed that the developed numerical model could predict the flexural behavior of beams with hybrid reinforcement with reasonable accuracy. Based on the test results, parametric analysis, and data obtained from the literature, the use of the neutral axis angle and displacement index value to evaluate the ductility of cross-sections with hybrid reinforcement is proposed.

## 1. Introduction

The main rationale for using hybrid Fiber Reinforced Polymer (FRP)–steel reinforcement in reinforced concrete structures is to compensate for the weaknesses of FRP, which is brittle, has a low elastic modulus [1], and is not resistant to fire [2,3,4,5]. Several researchers carried out studies on the flexural performance of reinforced concrete beams with hybrid FRP–steel reinforcement over the last 20 years [1,2]. With the increasing use of FRP in reinforced concrete construction, the number of studies investigating this hybrid reinforcement is also increasing [1,2,6,7,8,9,10,11,12,13,14,15,16,17,18]. Some research results related to these studies are described below.

Qu et al. [6] demonstrated that steel reinforcement combined with Glass Fiber Reinforced Polymer (GFRP) reinforcement increased the flexural capacity and ductility of reinforced concrete beams due to the presence of steel bars. Another study demonstrated that the ductility of beams with FRP reinforcement could be improved by either increasing the reinforcement ratio or adding conventional steel bars [7]. The arrangement of the steel bars played an essential role in the stiffness and crack width of reinforced concrete with hybrid FRP–steel reinforcement [8]. It was also reported that the deflection and crack width of beams with both hybrid Basalt Fiber Reinforced Polymer (BFRP) and steel reinforcement were smaller than those of beams reinforced with only BFRP bars [10].

Experimental tests using two types of FRP indicated that beams with hybrid GFRP–steel reinforcement demonstrated significantly reduced stiffness after the first cracking and yielding of steel reinforcement compared to hybrid beams with Carbon Fiber Reinforced Polymer (CFRP)–steel reinforcement [11]. A numerical study using cross-section analysis of reinforced concrete beams was also presented in the above reference. The analytical model showed promising results in predicting the moment capacity of reinforced concrete beams with hybrid Near Surface Mounted (NSM) FRP bar/strip reinforcement.

A study carried out by Refai et al. [12] concluded that hybrid reinforcement (steel and GFRP) improved the flexural behavior of the beam in terms of deformability, cracking, stiffness, and beam capacity compared to beams that only used GFRP reinforcement. In addition, to ensure adequate beam deformation, this paper recommended that beams with hybrid reinforcement design consider the yield of steel before concrete crushing or FRP fracture.

Qin et al. [14] carried out a numerical analysis using 15 finite element models. Six of them closely modeled experimental results for the flexural behavior of beams with hybrid FRP–steel reinforcement. It was concluded that the finite element model could be applied to predict the flexural behavior of hybrid FRP–steel beams accurately. This study pointed out that in over-reinforced beams, the hybrid reinforcement ratio, *A_f_*/*A_s_*, must be designed in the range of 1–2.5 to ensure that the beam retains strength with sufficient ductility and stiffness after exceeding the elastic phase.

In another study, Bui et al. [15] found differences in behavior between FRP–steel hybrid beams and conventional reinforced concrete beams. This paper showed that in beams with hybrid reinforcement, FRP reinforcement is responsible for ultimate capacity, while steel reinforcement is responsible for the ductility of the beam. Therefore, the ductility of a beam with a hybrid reinforcement can be increased if the *A_f_*/*A_s_* reinforcement ratio is small.

Araba and Ashour [16] demonstrated that the moment capacity of beams with hybrid reinforcement is more influenced by the ratio of hybrid reinforcement than the axial stiffness ratio (*A_s_E_s_*/*A_f_E_f_*). They also reported that the addition of axial stiffness is not proportional to the increase in moment capacity.

Sun et al. [17] experimentally examined the effect of reinforcement arrangement in beams with hybrid reinforcement. They discovered that the deflection of the beam with a distributed reinforcement arrangement was smaller than if the reinforcement was bundled. This study also showed that beams with hybrid reinforcement had sufficient ductility.

A paper presented by Kartal et al. [18] provided information concerning the zoning of the load–deflection curve. The load–deflection curve of beams with hybrid reinforcement could be divided into three parts: the regions before and after flexural cracking and the region after the tensile reinforcement yielded. This reference also stated that the deformation capacity of the beam with hybrid reinforcement increases with the increase in the amount of FRP reinforcement in the tension zone.

Recently, a parametric study was carried out to investigate the behavior of neutral axis growth in reinforced concrete sections with conventional steel, CFRP, and GFRP bars [19]. This research showed that the neutral axis curve in reinforced concrete cross-sections with steel bars could similarly be divided into three regions: before cracking, after cracking, and after the tensile reinforcement yielded. Meanwhile, in the cross-sections of reinforced concrete with FRP bars, only two regions existed: before and after cracking.

Current research conclusively demonstrates that beams with hybrid reinforcement have increased stiffness, flexural capacity, and ductility compared to beams reinforced with pure FRP. The results of previous studies also show that as the hybrid reinforcement ratio (*A_f_*/*A_s_*) increases, the determination of the ductility becomes more complex because the yield position of the steel is not clear. Although previous researchers have investigated many attributes of the flexural behavior of hybrid reinforced beams, some features of the performance of these beams are still unclear. In particular, the effects of the hybrid reinforcement ratio (*A_f_*/*A_s_*) and the arrangement of the hybrid reinforcements on the flexural performance of reinforced concrete beams with hybrid FRP–steel bars have not been thoroughly investigated.

Therefore, the purpose of this experimental study was to investigate the effects of the reinforcement ratio and the arrangement of hybrid reinforcement on the flexural performance and ductility of reinforced concrete beams with hybrid FRP–steel bars. Neutral axis angle and displacement index were also introduced to evaluate the ductility of beams with hybrid reinforcement. Curves of neutral axis growth obtained with the help of a computer program were used to determine the neutral axis angle and displacement index. This study also provided new data regarding the effects of using hybrid FRP–steel as internal reinforcement.

Fourteen reinforced concrete beams, consisting of six control beams and eight with hybrid reinforcement, were tested experimentally. The effect of the hybrid reinforcement ratio was examined by varying the amount of FRP and steel reinforcement in the tensile region of the beam. The influence of the reinforcement position was investigated using two types of bar arrangement. The experimental data in this study were combined with data obtained from parametric analysis and from the literature. Numerical analysis, to obtain flexural capacity, strain behavior, and neutral axis growth due to the effects of the test variables, was also performed using the fiber element model.

## 2. Experimental Study

The beams tested in this study were simply supported concrete beams. The beams were monotonically loaded until failure with a two-point load using a hydraulic jack. The loading position and dimensions of the beam are shown in Figure 1a. The clear span was 2000 mm for all tested beams, the shear span length was 800 mm, and the end anchorage length beyond the support was 150 mm. The dimensions of the beam cross-sections, the configurations of the reinforcement positions, and the types of reinforcement used are shown in Figure 1b–o. The thickness of the concrete cover was 30 mm.

Two types of cross-section, as shown in Figure 1, were used. The first type (Type I) was a beam section with one layer of tensile reinforcement (Figure 1b), and the second type (Type II) was a beam section with two layers of tensile reinforcement (Figure 1c). The six control beams consisted of two beams with steel reinforcement (BFS-2 and BFS-4), two beams with GFRP reinforcement (BFG-1 and BFG-2), and two beams with CFRP reinforcement (BFC-1 and BFC-2). The eight beams with FRP–steel hybrid reinforcement were distinguished by the two types of cross-section and the ratio of FRP reinforcement to steel reinforcement.

Figure 2a shows the FRP reinforcement used in this study. Reinforcement cages before concrete casting and experimental setup and equipment used in beam tests are shown in Figure 2b,c, respectively. The longitudinal reinforcement used consisted of steel bars with a diameter of 13 mm and a yield stress (*f_y_*) of 375 MPa, GFRP bars with a diameter of 13 mm, an ultimate tensile strength (*f_fu_*) of 788 MPa, and a modulus of elasticity (*E_f_*) of 43.9 GPa, and CFRP bars with a diameter of 13 mm, a maximum tensile strength (*f_fu_*) of 2070 MPa, and a modulus of elasticity (*E_f_*) of 124 GPa. The ratios of FRP reinforcement to steel reinforcement were 0.5 and 2.0. The GFRP and CFRP rods used, shown in Figure 2a, were supplied by FYFE Co. LLC from the USA. The mechanical properties of the FRP rods used in this study were obtained from the leaflet issued by the manufacturer. Transverse reinforcements with a diameter of 10 mm and yield stress of 454 MPa were used for all specimens. The properties of the materials used in this study as determined via experimental studies, parametric studies, and data adopted from the literature are shown in Table 1. Fresh concrete was ordered from a ready-mix company and the compressive strength (*f_c_*’) of the concrete at 28 days was 20 MPa.

Load was measured using load cells in the experimental study, and deflection was measured using linear variable displacement transducers (LVDTs). The load cells and LVDTs were connected to a data acquisition system, and data were collected on a data storage system. Load was increased gradually until failure occurred; the control mode used for the beam test was load control. The test setup, load position, and LVDT placement on the test beam are shown in Figure 2c. The load cells, LVDTs, and data logger used were products of Tokyo Measuring Instruments Laboratory Co., Ltd., Tokyo, Japan.

Additional data were also obtained from the literature [1,6,7,8,12] to increase the data population with further variations in the hybrid reinforcement ratio, ranging from 0.3 to 2.9. Two additional types of FRP materials (Aramid Fiber Reinforced Polymer (AFRP) and GFRP) were represented in these experimental data adopted from the literature, as are two additional types of cross-sections based on the number of layers of tensile reinforcement.

## 3. Analytical Study

The analytical study used a fiber element model, as shown in Figure 3, to obtain the theoretical moment–curvature curve of the reinforced concrete sections using a nonlinear material stress–strain relationship [20]. The theoretical moment–curvature curves for reinforced concrete cross-sections under flexural loads can be derived based on the following assumptions: the cross-sections before bending remain plane after bending, and the stress–strain curves for concrete and steel are known. Moment–curvature analysis is also a method for accurately determining a reinforced concrete section’s load–deformation behavior.

This method begins by dividing the reinforced concrete section into layers of small elements. The reinforcement is positioned in layers parallel to the concrete elements, and the reinforcement distance is measured from the top fiber of the cross-section. Strain in each layer (*ε_i_*) can then be calculated using the distance from each element to the top of the cross-section (*y_i_*) and by assuming curvature (*μ*) (Equation (1)).
(1)$$\varepsilon_{i}=\varepsilon_{o}-\left(\mu y_{i}\right)$$

The force on each element (*F_i_*) is calculated using the stress on (*σ_i_*) and the area of (*A_i_*) each element. The stress is obtained from the nonlinear material stress–strain relationship inputted in the previous step. The material stress–strain models used in this study are shown in Figure 4.
(2)$$\sigma_{i}=f\left(\varepsilon_{i}\right)$$

The stress–strain relationship of concrete in compression used in this study was adopted from the model proposed by Mander et al. [21], as shown in Figure 4a. For concrete in tension, a linear model up to the maximum tensile strength of the concrete without tensile stiffness was used. The tensile strength of the concrete was considered in the flexural analysis to obtain the pre-cracking regions of the moment–curvature and neutral axis curves.

The stress–strain relationship for the steel bar used in this study was a bilinear model, while a linear model up to failure was used for CFRP and GFRP bars, as shown in Figure 4b,c.
(3)$$F_{i}=A_{i}\sigma_{i}$$

The equilibrium condition is found by adding all the internal forces. If the equilibrium conditions are not satisfied, the calculation will return to the previous process by changing the strain (*ε_o_*) on the centroid axis. After the equilibrium condition is met, the moment (*M*) at each load step is obtained by multiplying the obtained internal forces by the distance from each element to the top of the cross-section (Equation (4)).
(4)$$M=\Sigma F_{i}y_{i}$$

The deflection can be calculated using the curvature at each step. The neutral axis depth (*c*) at each step can be calculated using Equation (5). The analytical calculation process described was assisted by a computer program developed by the author, and the algorithm for this calculation process can be found in the literature [22,23,24,25,26].
(5)$$c=\frac{\varepsilon_{cm}}{\mu }$$

The parametric study was carried out to fill in the gaps in the hybrid reinforcement ratio (*A_f_*/*A_s_*) data that could not be determined from experimental studies or the literature. There were 12 specimens in the parametric study, with hybrid reinforcement ratios ranging from 1.3 to 4.3. With these data, it was expected that the behavior of cross-sections with higher hybrid reinforcement ratios could be represented. Based on the number of layers of tensile reinforcement, the beam cross-sections in the parametric study consisted of the same two types as in the experimental research, namely, Type I and Type II. This parametric study used two FRP materials (GFRP and CFRP).

## 4. Results and Discussion

### 4.1. Crack Patterns and Failure Modes of the Tested Beams

The crack pattern and failure model for each beam are shown in Figure 5. All beams experienced flexural failure, indicated by concrete crushing at the top of the compression zone. The first flexural crack occurred in the constant moment zone, with an average value of 4.5 kN. This value is relatively similar to the average value of the first crack obtained from the cross-sectional analysis, which was 3.8 kN. The first crack load value of each specimen is shown in Table 2.

It is shown in Figure 5 that the height of the flexural crack in the constant moment zone varied relative to the hybrid reinforcement ratio (*A_f_*/*A_s_*). The higher the hybrid reinforcement ratio, the lower the flexural cracking in the constant moment zone, as shown in Figure 5j,n. As the load increased, flexural cracks spread to the shear span zone, developing into shear cracks. Inclined cracks were dominant in beams with higher hybrid reinforcement ratios. When inclined cracks are formed and propagate towards the load position, the stress at the top of the compression zone increases until the beams reach their failure condition. A summary of the failure modes of the tested beams is shown in Table 2. Different reinforcing materials (GFRP and CFRP) caused significant differences in crack patterns. Cracks in beams with GFRP reinforcement were wider and higher than beams with CFRP reinforcement, due to the lower modulus of elasticity of GFRP.

A balanced reinforcement ratio of the cross-section was evaluated based on the equation suggested in ACI 440.1R-15 [27], expressed in Equation (6).
(6)$$\rho_{bf}=0.85\;\beta_{1}\frac{fc^{\prime }}{f_{fu}}\;\frac{E_{f}\varepsilon_{cu}}{E_{f}\varepsilon_{cu}+f_{fu}}$$

The effective ratio of hybrid reinforcement is calculated by the following Equation (7). The reinforcement ratio for beams with hybrid reinforcement can be calculated using Equations (6) and (7). In this research, steel bars in cross-sections with hybrid reinforcement were designed to yield and FRP reinforcement still in elastic conditions without rupturing. The calculated reinforcement ratios of the tested specimens are shown in Table 2.
(7)$$\rho_{eq}=\rho_{s\;}\frac{f_{y}}{f_{fu}}+\rho_{f}$$

### 4.2. Effect of Longitudinal Hybrid Reinforcement Ratio (A_f_/A_s_)

A load–deflection curve showing the capacity of each beam is plotted in Figure 6. Only the output of the LVDT at the mid-span of the beams is plotted in Figure 6 because the greatest deflection occurs at the mid-span of the beam. A comparison of the capacities of the non-hybrid beams is shown in Figure 6a,b. These figures show that behavior differed between steel- and FRP-reinforced beams. Beams with steel reinforcement demonstrated ductile behavior, while FRP-reinforced beams did not. The ductility of beams with steel reinforcement can be seen from the decrease in tensile reinforcement, which is characterized by a sudden change in stiffness above the elastic limit (after the tensile reinforcement yields) without any significant change in resistance capacity.

The behavior of the load–deflection curves of beams with hybrid reinforcement can be seen in Figure 6c–j. The ductility of beams with hybrid reinforcement was highly dependent on the ratio of hybrid reinforcement (*A_f_*/*A_s_*). The highest ductility was seen in beams with a small hybrid reinforcement ratio. By contrast, ductility decreased with a higher hybrid reinforcement ratio, even though the steel yielded.

Figure 6c–j also show that in beams with a higher hybrid reinforcement ratio (BHG-4 and BHC-4), the yield point of the reinforcing steel was not visible. This caused the process of calculating beam ductility to become problematic.

### 4.3. Effect of Reinforcement Position

The effects of the position of the reinforcement on the tested beam are shown in Figure 7. It can be seen that the position of the reinforcement slightly affected the capacity and ductility of the beam. Beams with Type I reinforcement (one layer of reinforcement) showed higher capacity but slightly lower ductility.

Figure 7 also shows that the position of the reinforcement and the hybrid reinforcement ratio affected the slope of the load–deflection curve in the post-elastic region. This difference is seen in Figure 7d–g. The difference in slope in the post-elastic region due to the position of the reinforcement is visible in the beams with a hybrid reinforcement ratio of 0.5 by comparing Figure 7d,f. However, the beams with hybrid reinforcement ratios of 2 did not show a significant difference in the slope of the load–deflection curve in the post-elastic region.

The difference in flexural stiffness due to the effect of the longitudinal hybrid reinforcement ratio (*A_f_*/*A_s_*) and the effect of the position of the reinforcement can be seen in Table 2 and plotted in Figure 8. This study used the secant method to calculate the flexural stiffness from the load–deflection curve before yielding. It can be seen from Figure 8 that cross-sections with Type I reinforcement positions showed higher stiffness values than those with Type II. In addition, a smaller hybrid reinforcement ratio resulted in a higher stiffness value. Furthermore, cross-sections with hybrid reinforcement using CFRP exhibited higher stiffness values than those using GFRP.

### 4.4. Strain Distribution in Cross-Sections with Hybrid Reinforcement

The strain distributions of the tested beams, obtained from analytical calculation using Equation (1), are shown in Figure 9. The notation *y* on the vertical axis represents the height of each element measured from the top of the cross-section, and the notation *H* represents the height of the section. It can be seen from the figure that yielding occurred in cross-sections with steel reinforcement because the strain that occurred in the steel reinforcement (*ε_s_*) exceeded the yield strain of steel (*ε_y_*).

The maximum strain in the FRP reinforcement (*ε_f_*) in all beam sections did not exceed the ultimate tensile strength of the material (*ε_fu_*). This result agrees with the beam test results obtained from experiments. In cross-sectional analysis via the fiber element method, the maximum compressive strain entered into the computer program was 0.003, which agreed with the experimental results, where all beams were crushed in the compression zone.

In cross-sections with steel reinforcement, the maximum strain in Type I sections was greater than that in Type II sections. However, this did not happen in FRP-reinforced and hybrid-reinforced sections. This indicates that the position of the reinforcement affected the cross-sectional strain distribution. The different types of FRP also had a significant influence on the strain distribution, where cross-sections with GFRP reinforcement experienced higher strain. This is due to the lower modulus of elasticity of GFRP compared to steel and CFRP.

### 4.5. Neutral Axis Growth in Cross-Sections with Hybrid Reinforcement

As stated in the previous section, the process of calculating the ductility from the load–deflection curve becomes more difficult as the hybrid reinforcement ratio increases, because the exact point at which the steel reinforcement yields becomes unclear. In this study, the authors propose a method for determining the yield point of steel reinforcement in a reinforced concrete beam cross-section with hybrid reinforcement. The proposed method uses an analytically obtained neutral axis curve, calculated using Equation (5). Before being applied for this purpose, the calculated neutral axis curve results were verified using the results of other software calculations [28,29].

Figure 10 and Figure 11 compare the neutral axis curves calculated with Equation (5) using the results of other softwares. The vertical axis represents the normalized neutral axis height (*c*) with the effective depth (*d*), and the horizontal axis represents the moment value at each load step (*M*) normalized to the ultimate moment value (*M_u_*). These comparisons show that the neutral axis curves calculated with Equation (5) are very close to the results obtained using other softwares. This proves that a neutral axis curve calculated with Equation (5) can be used to determine the yield point of the steel reinforcement in a reinforced concrete cross-section with hybrid reinforcement.

Figure 12 shows the neutral axis curves obtained analytically from the tested reinforced concrete beam sections. In all beam cross-sections, the movement of the neutral axis starts with the segment from Point *A* to Point *B*. This represents the region before the first crack in the section. After that, the neutral axis moves towards Point *C*, which is a point that indicates that the reinforcing steel has yielded. In cross-sections with steel reinforcement (BFS-2 and BFS-4), Point *C* is very clearly visible, followed by a steep slope of inclination towards Point *D*, which indicates movement after the elastic region of the steel.

On the other hand, Point *C* (the yield point) does not exist in the cross-sections with only FRP reinforcement (BFC-1, BFC-2, BFG-1, and BFG-2). In these cross-sections, the neutral axis moves towards the ultimate point (*D*) without any post-elastic behavior of the reinforcement after Point *B*.

In cross-sections with hybrid reinforcement, Point *C* is followed by a post-elastic region where the angle of inclination depends on the hybrid reinforcement ratio. The larger the hybrid reinforcement ratio, the smaller the angle of inclination of the curve after Point *C*. Therefore, Point *C* is used as a reference to help determine the yield point of sections with large hybrid reinforcement ratios.

In general, the movement of the neutral axis curve can be summarized in Figure 13. The neutral axis curves of specimens BFS-2, BFC-1, BFG-1, BHG-1, and BHC-1 are plotted to demonstrate the regions and essential points. It can be seen that the starting point of the neutral axis is at approximately the middle of the cross-sectional height (Point *A*). The movement of the neutral axis does not show a significant change until the cross-section experiences the first flexural crack (Point *B*). After flexural cracking occurs, the neutral axis moves rapidly towards the top of the cross-section.

In cross-sections with steel reinforcement, the neutral axis moves to Point *C*, representing the steel’s yield point. In an over-reinforced cross-section with steel bars, the tensile reinforcement in the cross-section does not experience yielding. After point *B,* the curve goes straight towards, and stops at, Point *D*. This phenomenon is also seen in all neutral axis curves in cross-sections with FRP reinforcement, indicating that the inelastic deformation of the steel reinforcement affects the movement of the neutral axis and the ductility of the cross-section.

During loading, the curve of the neutral axis can be divided into three regions, as shown in Figure 13a, especially in sections under-reinforced with steel. However, with an over-reinforced cross-section (where the reinforcement does not yield), there are only two segments to the curve, with the end of the curve displaying a downward slope before reaching the ultimate condition. In this condition, the section fails in compression, characterized by crushing of the concrete in the compression area with the reinforcement still in an elastic state. A similar situation occurs in sections under- or over-reinforced with FRP, as shown in Figure 13b.

The behavior of the neutral axis curve in a concrete cross-section with hybrid reinforcement is shown in Figure 13c,d. There are three regions on these curves, namely the region before cracking (*A* to *B*), the region after cracking (*B* to *C*), and the region after yielding (*C* to *D*). The angle of inclination (*α*) after Point *C* depends on the type of FRP used and the hybrid reinforcement ratio.

### 4.6. Ductility of Cross-Sections with Hybrid Reinforcement

The process of calculating the ductility of beams with and without hybrid reinforcement was carried out using the load–deflection curve shown in Figure 14 and Equation (8). Because it was difficult to determine the position of the yield load (*P_y_*) visually when the hybrid reinforcement ratio was large, the positions of *P_y_* for beam cross-sections with large hybrid reinforcement ratios (BHG-2, BHG-4, BHC-2, and BHC-4) were determined using the neutral axis curves shown in Figure 12 above. The yield load positions for cross-sections with large hybrid reinforcement ratios correspond to point *C* on the neutral axis curve. The calculated ductility values are presented in Table 3.
(8)$$\delta =\frac{\Delta_{u}}{\Delta_{y}}$$

Figure 15 shows the effect of the hybrid reinforcement ratio (*A_f_*/*A_s_*) on the ductility (*δ*) of the tested beams, along with comparable data from the parametric study and obtained from the literature. The hybrid reinforcement ratios used in the specimens for experimental and parametric testing and derived from data obtained from the literature ranged from 0.3 to 4.3, with three types of FRP bars (AFRP, CFRP, and GRP). The red line shows the data trend, which indicated a decrease in ductility with increases in the hybrid reinforcement ratio. The vertical shaded area in Figure 15 indicates the limitation of the hybrid reinforcement ratio (1 to 2.5) recommended to ensure that the beam retains sufficient ductility and stiffness after exceeding the elastic phase [14].

Statistical analysis was used to find the correlation coefficient between the hybrid reinforcement ratio (*A_f_*/*A_s_*) and the ductility (*δ*). The results showed a moderate correlation with an *R* of −0.38, where the minus sign means that the value of the hybrid reinforcement ratio increases as the ductility value decreases.

### 4.7. Effect of FRP Type

The effect of the FRP reinforcement type (GFRP and CFRP) used for hybrid reinforcement of cross-sections was very significant. This is due to the different elastic moduli of GFRP and CFRP. From the results presented above, it can be seen that the deflection of the hybrid specimen with GFRP was greater than that of the hybrid specimen with CFRP. A larger deflection causes more cracks, higher cracks, and larger crack widths. Higher cracks can be seen from a comparison of the neutral axis curves of the two types of material, shown in Figure 16a (specimens BHC-1 and BHG-1). Despite its larger capacity, the hybrid specimen reinforced with CFRP had lower ductility, as shown in Figure 16b.

### 4.8. Neutral Axis Angle (α) and Displacement Index (δ_N_)

The neutral axis angle (*α*) and displacement index (*δ_N_*) obtained from the neutral axis curve profile were introduced as an alternative method for evaluating the ductility of concrete sections with hybrid reinforcement. Figure 17a,b illustrate the moment–curvature curve and the corresponding neutral axis curve obtained from cross-section analysis. The definition of the neutral axis angle in this paper was the angle formed at Point *C* (see Figure 17b), following the yield point of the steel reinforcement. The neutral axis angle can be calculated using Equation (9) and with the help of Figure 17b. The value of this angle indicates the change in stiffness of the section after the yield point of the steel reinforcement. A positive angle value means that Point *D* is above Point *E* and vice versa. The greater the neutral axis angle value, the greater the ductility value.

The displacement index (*δ_N_*) value can be calculated using Equation (10). This value denotes the propagation of the neutral axis from Point *A* to Point *D*. An index value greater than one indicates that Point *D* is above Point *E*, whereas an index value smaller than one means that Point *D* is below Point *E*. The greater the displacement index value, the greater the ductility value. Neutral axis angles and displacement index values calculated from experimental data, parametric studies, and the literature are shown in Table 3 and Figure 18 and Figure 19.
(9)$$\tan\alpha =\frac{y_{E}-y_{D}}{x_{E}-x_{C}}$$
(10)$$\delta_{N}=\frac{y_{A}-y_{D}}{y_{A}-y_{E}}$$

The neutral axis angle (*α*) is plotted against the hybrid reinforcement ratio (*A_f_*/*A_s_*) in Figure 18. It can be seen that the neutral axis angle decreased with increases in the hybrid reinforcement ratio. This result is in accordance with the observation in the previous section that ductility decreases with increases in the hybrid reinforcement ratio.

The neutral axis angle data plotted in Figure 18 are combined in Figure 19a. The red line shows the data trend, and indicates a significant negative correlation value (*R*) of −0.67. Figure 19b shows the relationship between the hybrid reinforcement ratio (*A_f_*/*A_s_*) and the neutral axis displacement index (*δ_N_*). The data have a relatively strong negative correlation value of −0.83: the value of the neutral axis displacement index decreased as the hybrid reinforcement ratio increased.

The neutral axis angles and displacement index values are plotted in Figure 19c to show the correlation between the two variables. The data have a relatively strong positive correlation value of 0.89, indicating that these two variables had a reasonably strong relationship.

Furthermore, the relationship between the neutral axis displacement index value and ductility is shown in Figure 19d. The data show a weak positive linear relationship with a correlation value of 0.26. Nevertheless, these data indicate that the ductility value could increase with the neutral axis displacement index value.

Refai et al. [12] proposed the use of a modified deformation factor to measure the ability of a beam to sustain inelastic deformation and significant rotation before failure. This is the ratio of the product of the moment and the curvature at the ultimate yield point to the product of the moment and the curvature at the yield point of the steel reinforcement, and is given in Equation (11). The calculated deformation factors (*DF*) for all data in this study are shown in Table 3.
(11)$$DF=\frac{M_{u}\mu_{u}}{M_{y}\mu_{y}}$$

The relationships between the deformation factor and the reinforcement ratio (*A_f_*/*A_s_*), neutral axis angle (*α*), neutral axis displacement index value (*δ_N_*), and ductility (*δ*) are shown in Figure 20, with correlation values of −0.1, −0.06, 0.06, and 0.58 respectively.

The trend lines in Figure 20a,b show that a decrease in the deformation factor correlated with increases in the reinforcement ratio and neutral axis angle. Figure 20c,d show that an increase in the deformation factor correlated with increases in neutral axis displacement index value and ductility. Figure 20d indicates a significant correlation between the deformation factor and ductility.

The low correlation values are probably due to the above analysis using data from the literature that is based on varied material types (e.g., FRPs and steels with different moduli of elasticity and tensile strengths; differences in concrete quality, cross-sectional size, or beam dimensions). Therefore, in Figure 21, only the parametric study data are presented. In these parametric studies, only the hybrid reinforcement ratio variable was varied, as shown in Table 2 above.

It can be seen from these figures that the correlation was strong between each pair of factors (with absolute values ≥ 0.65). Using the results presented in Figure 21, we can determine the recommended values of parameters *α*, *δ_N_*, *δ*, and *DF* for cross-sections with hybrid reinforcement. The dotted circle indicates the range of recommended values for an over-reinforced cross-section with hybrid reinforcement (1–2.5).

As all the reinforcement ratios in the hybrid cross-section specimens are categorized as over-reinforced, the process of determining the recommended values of the parameter begins by using the suggested limitation value for the hybrid reinforcement ratio [14], which is the range of 1–2.5, as shown in Figure 21a. The zone where the α parameter value meets the recommended range (1–2.5) is >0°. This process is repeated to obtain the other parameter limit values (Figure 21b–f). The resulting displacement index (*δ_N_*) and ductility (*δ*) values that meet the recommended range are >1 and >4, respectively, while the deformation factor (*DF*) value that meets the recommended range is >6. A summary of the recommended values for *α*, *δ_N_*, *δ*, and *DF* in hybrid reinforced beams with adequate ductility is found in Table 4.

## 5. Conclusions

Fourteen reinforced concrete beams were tested to evaluate ductility, and then analyzed along with data from parametric studies and data obtained from the literature covering various hybrid reinforcement ratios and reinforcement positions. Some conclusions that can be drawn from this study are:(1).The reinforcement ratio (*A_f_*/*A_s_*) strongly influenced the capacity and cross-sectional ductility of hybrid reinforced concrete. The higher the hybrid reinforcement ratio, the greater the capacity, but at the cost of ductility, which decreased with increases in the hybrid reinforcement ratio.(2).The position of the reinforcement slightly affected the capacity and ductility of the beam. Differences in slope in the post-elastic area were seen in beams with lower hybrid reinforcement ratios. However, beams with higher hybrid reinforcement ratios did not show a significant difference in the slope of the load–deflection curve in the post-elastic region.(3).Cross-sections with Type I reinforcement positions showed higher flexural stiffness values than with Type II positions. A smaller hybrid reinforcement ratio resulted in a higher flexural stiffness value. Moreover, cross-sections with hybrid reinforcement using CFRP exhibited higher flexural stiffness values than those using GFRP.(4).The type of material used for FRP reinforcement (GFRP or CFRP) significantly affected the profile of the neutral axis curve, capacity, and ductility of hybrid reinforced cross-sections.(5).There were three regions on the neutral axis curve for cross-sections with hybrid reinforcement, i.e., the region before cracking, the region after cracking, and the region after yielding. The inclination of the neutral axis angle (*α*) after reinforcement yield depended on the type of FRP and the hybrid reinforcement ratio used.(6).The ductility of hybrid reinforced beams increased as the neutral axis angle increased. There were significant correlations between the neutral axis angle, neutral axis deformation index value, ductility, and deformation factor. The deformation factor increased with increasing neutral axis angles and deformation index values.(7).This result suggests that the neutral axis angle (*α*) and the deformation index value (*δ_N_*) proposed in this paper can be used to evaluate the ductility of cross-sections with hybrid reinforcement.

## Figures and Tables

**Figure 1 polymers-14-01017-f001:**
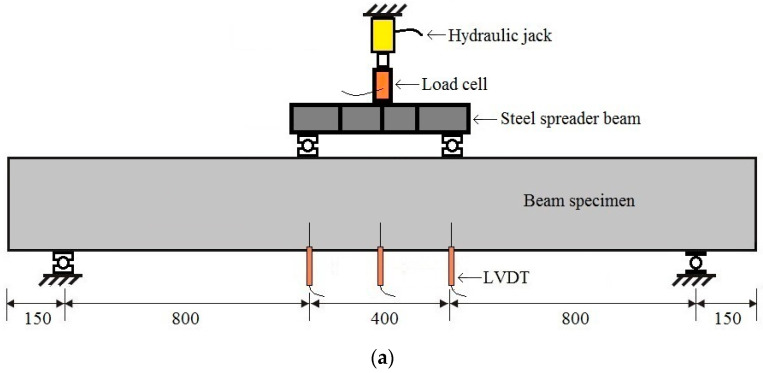
Schematic pictures of the tested beams and their identifications: (**a**) beam dimensions and loading position, (**b**) BFS-2, (**c**) BFS-4, (**d**) BFG-1, (**e**) BFG-2, (**f**) BFC-1, (**g**) BFC-2, (**h**) BHC-1, (**i**) BHC-2, (**j**) BHC-3, (**k**) BHC-4, (**l**) BHG-1, (**m**) BHG-2, (**n**) BHG-3, and (**o**) BHG-4.

**Figure 2 polymers-14-01017-f002:**
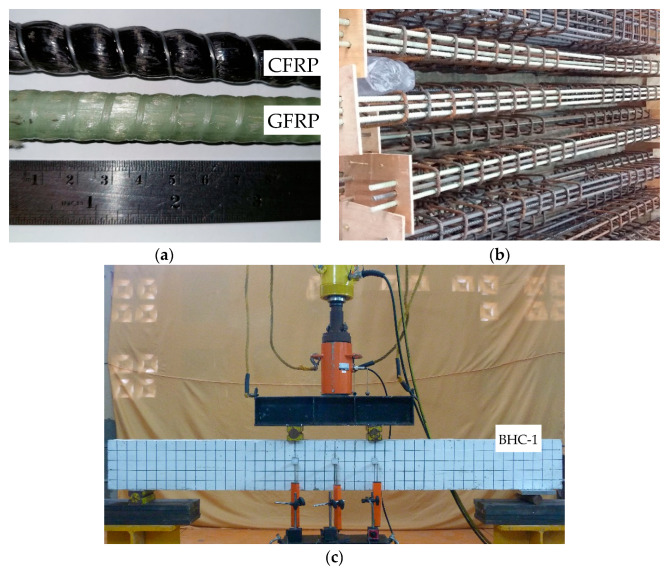
(**a**) GFRP and CFRP bars used in this study, (**b**) reinforcement cages before concrete casting, and (**c**) experimental setup and equipment used in beam test.

**Figure 3 polymers-14-01017-f003:**
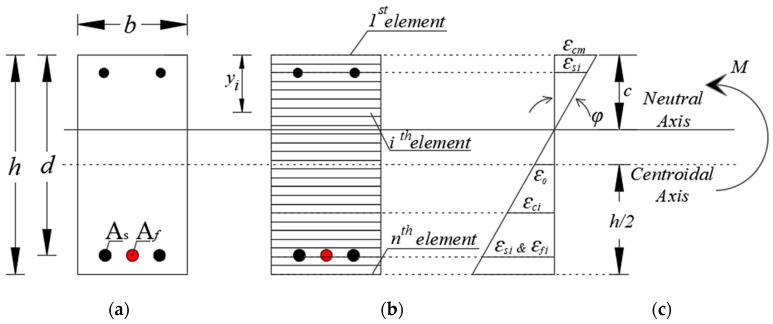
Analytical model using the fiber element model: (**a**) reinforced concrete cross-section, (**b**) fiber element model, and (**c**) strain distribution.

**Figure 4 polymers-14-01017-f004:**
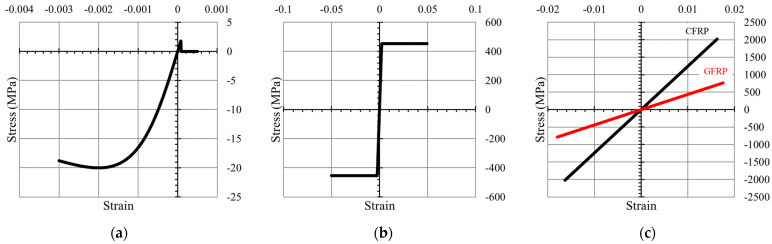
Stress–strain models used in fiber element analysis: (**a**) concrete, (**b**) bilinear model for steel reinforcement, and (**c**) linear model for CFRP and GFRP bars.

**Figure 5 polymers-14-01017-f005:**
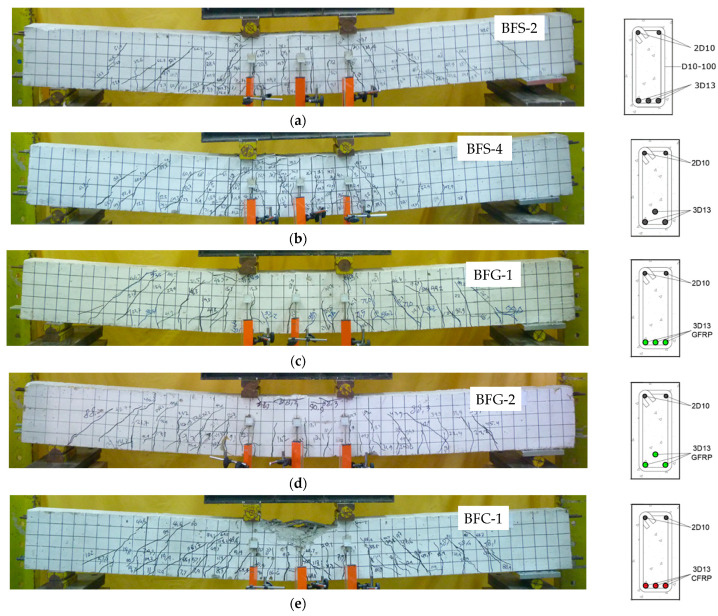
Crack patterns and failure modes of the tested beams: (**a**) BFS-2, (**b**) BFS-4, (**c**) BFG-1, (**d**) BFG-2, (**e**) BFC-1, (**f**) BFC-2, (**g**) BHG-1, (**h**) BHG-2, (**i**) BHG-3, (**j**) BHG-4, (**k**) BHC-1, (**l**) BHC-2, (**m**) BHC-3, and (**n**) BHC-4.

**Figure 6 polymers-14-01017-f006:**
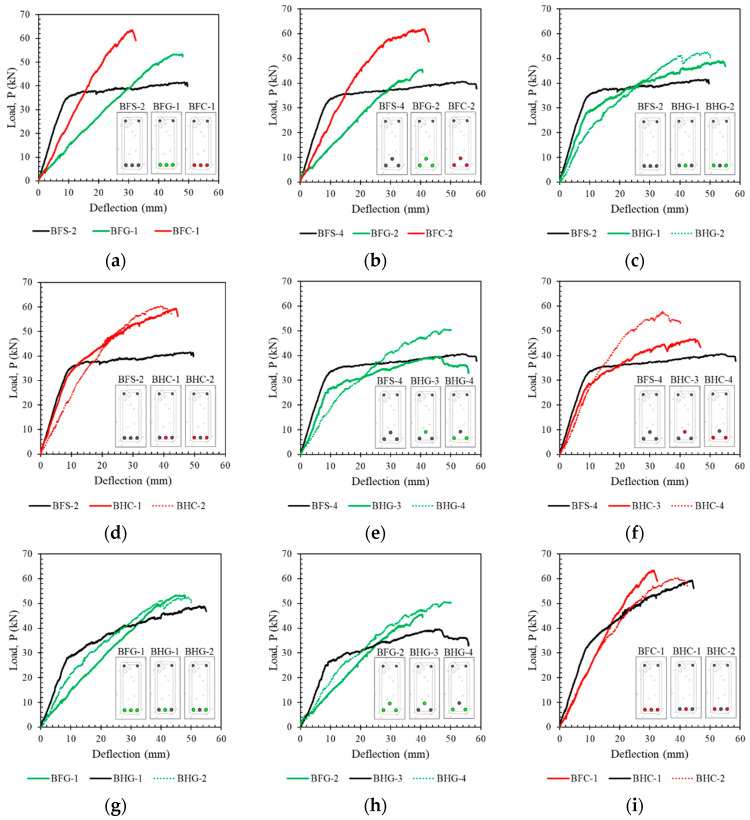
The load–deflection curves of the tested beams relative to the effect of the hybrid reinforcement ratio (**a**) BFS-2, BFG-1, and BFC-1, (**b**) BFS-4, BFG-2, and BFC-2, (**c**) BFS-2, BHG-1, and BHG-2, (**d**) BFS-2, BHC-1, and BHC-2, (**e**) BFS-4, BHG-3, and BHG-4, (**f**) BFS-4, BHC-3, and BHC-4, (**g**) BFG-1, BHG-1, and BHG-2, (**h**) BFG-2, BHG-3, and BHG-4, (**i**) BFC-1, BHC-1, and BHC-2, (**j**) BFC-2, BFC-3, and BHC-4.

**Figure 7 polymers-14-01017-f007:**
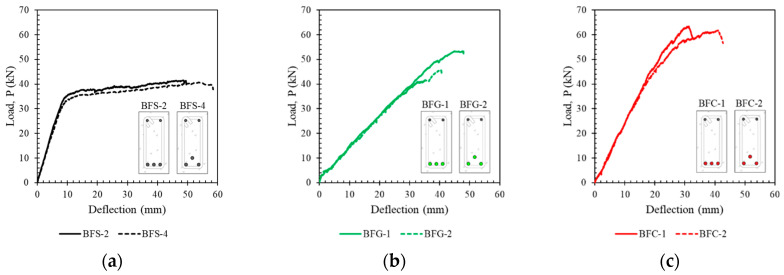
The load–deflection curves of the tested beams relative to the effect of the reinforcement position (**a**) BFS-2 and BFS-4, (**b**) BFG-1 and BFG-2, (**c**) BFC-1 and BFC-2, (**d**) BHG-1 and BHG-3, (**e**) BHG-2 and BHG-4, (**f**) BHC-1 and BHC-3, (**g**) BHC-2 and BHC-4.

**Figure 8 polymers-14-01017-f008:**
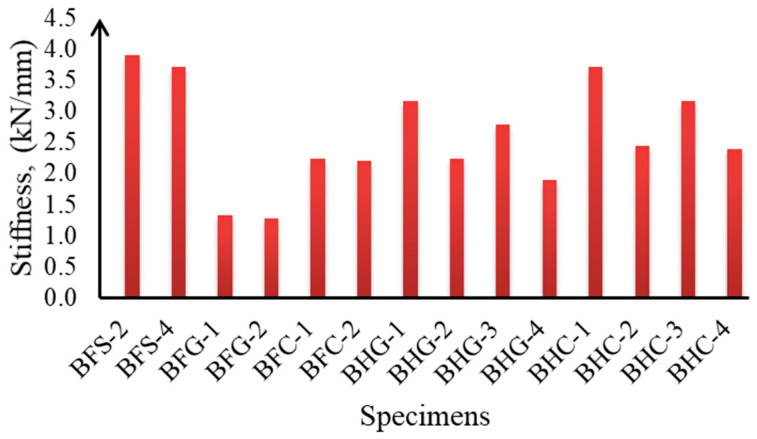
Flexural stiffness of the tested beams.

**Figure 9 polymers-14-01017-f009:**
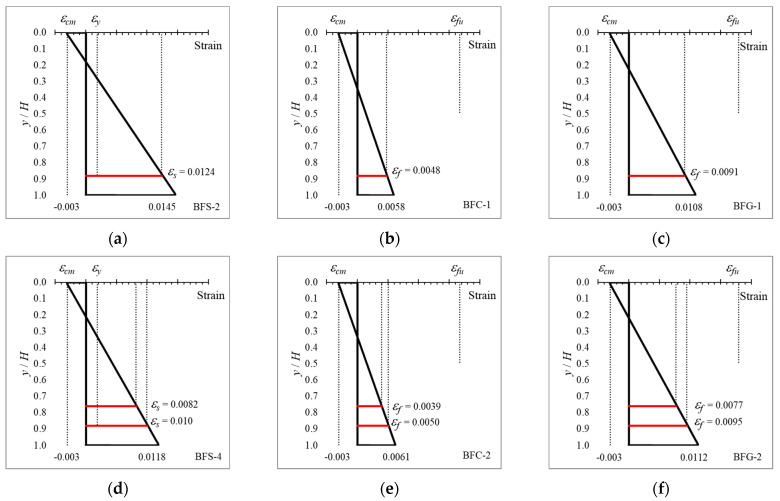
Strain distribution of the tested beams, obtained analytically: (**a**) BFS-2, (**b**) BFC-1, (**c**) BFG-1, (**d**) BFS-4, (**e**) BFC-2, (**f**) BFG-2, (**g**) BHG-1, (**h**) BHG-2, (**i**) BHG-3, (**j**) BHG-4, (**k**) BHC-1, (**l**) BHC-2, (**m**) BHC-3, and (**n**) BHC-4.

**Figure 10 polymers-14-01017-f010:**
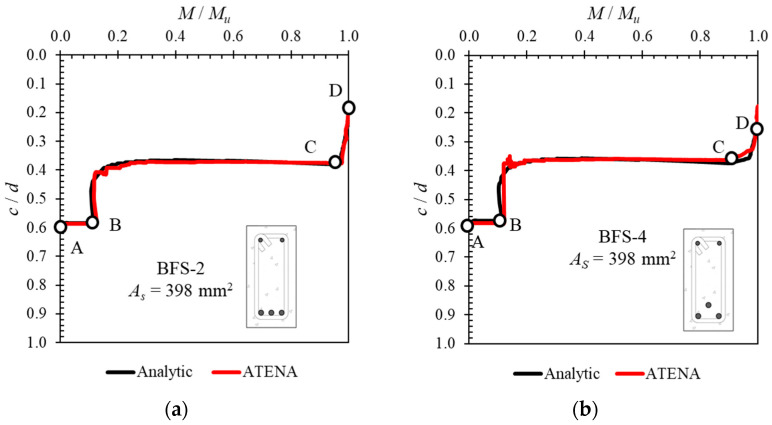
Verification of calculated neutral axis curve using results from ATENA software [28] for specimens: (**a**) BFS-2 and (**b**) BFS-4.

**Figure 11 polymers-14-01017-f011:**
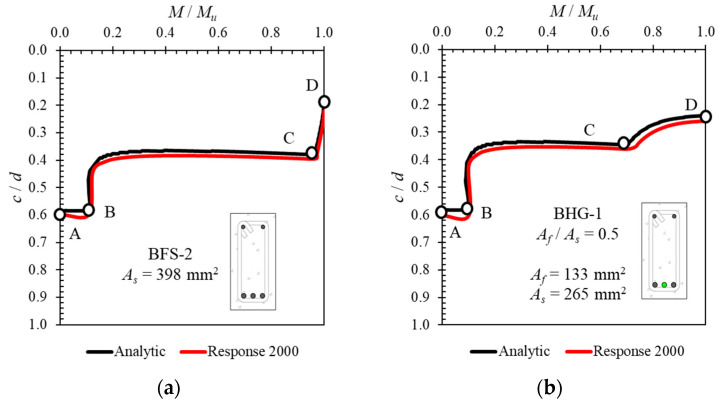
Verification of calculated neutral axis curve using results from Response 2000 software [29] for specimens (**a**) BFS-2 and (**b**) BHG-1.

**Figure 12 polymers-14-01017-f012:**
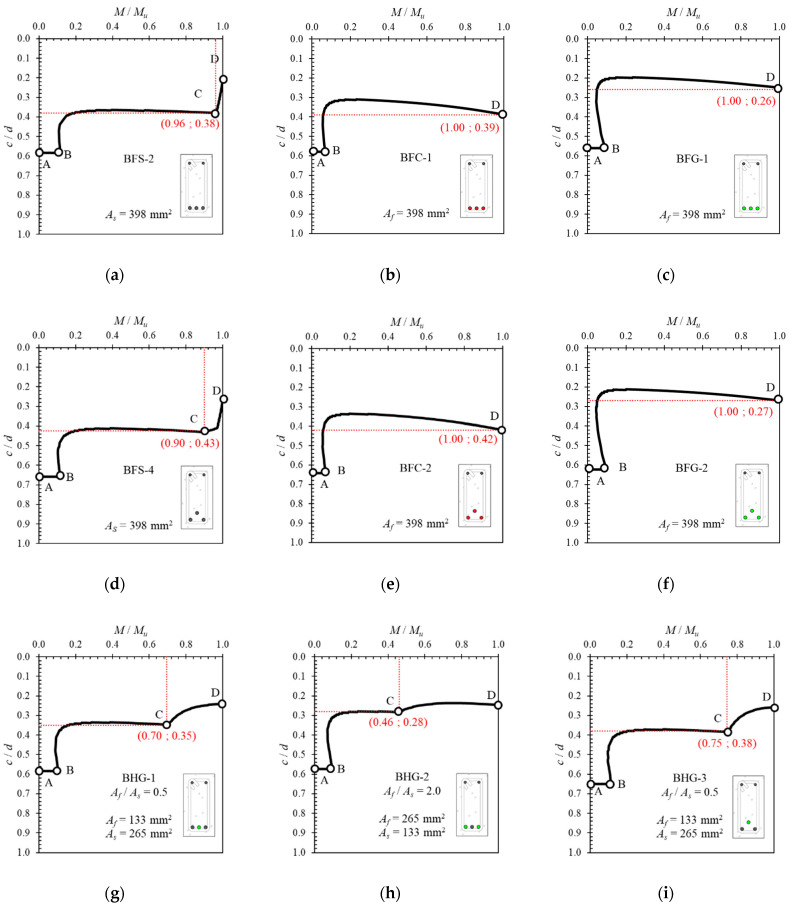
Neutral axis curves of the tested beams, obtained analytically: (**a**) BFS-2, (**b**) BFC-1, (**c**) BFG-1, (**d**) BFS-4, (**e**) BFC-2, (**f**) BFG-2, (**g**) BHG-1, (**h**) BHG-2, (**i**) BHG-3, (**j**) BHG-4, (**k**) BHC-1, (**l**) BHC-2, (**m**) BHC-3, and (**n**) BHC-4.

**Figure 13 polymers-14-01017-f013:**
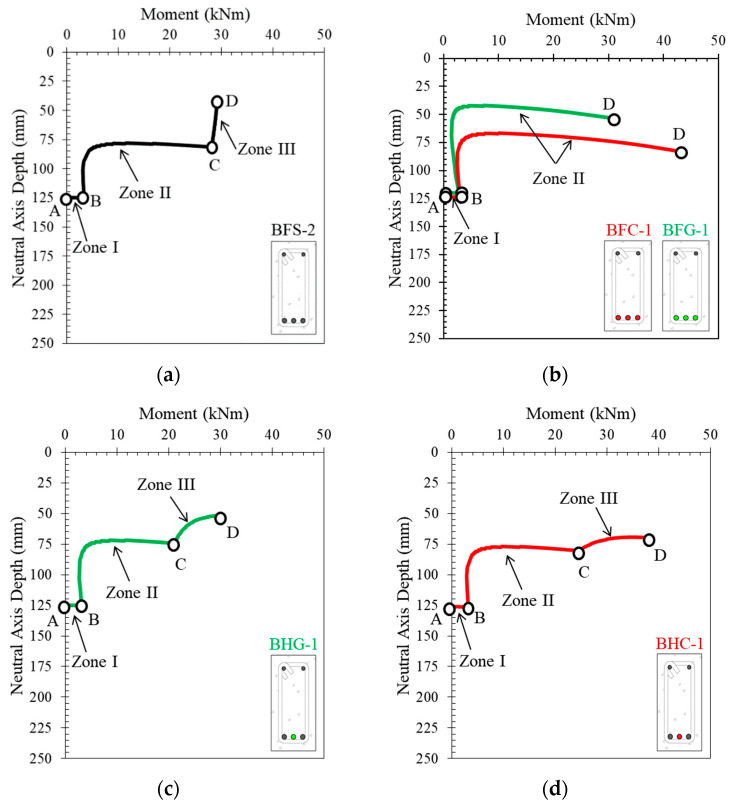
Typical neutral axis curves of reinforced concrete beam cross-sections with (**a**) steel reinforcement, (**b**) FRP reinforcement, (**c**) hybrid steel–GFRP reinforcement, and (**d**) hybrid steel–CFRP reinforcement.

**Figure 14 polymers-14-01017-f014:**
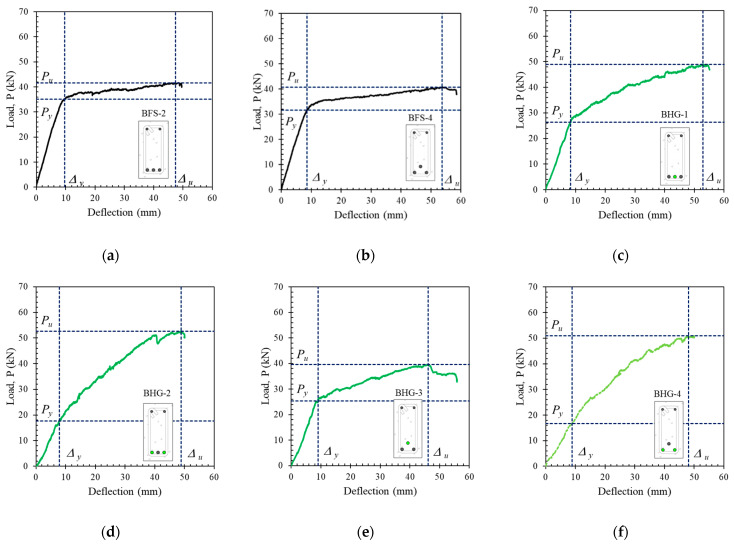
Ductility of tested beams determined using the load–deflection curve: (**a**) BFS-2, (**b**) BFS-4, (**c**) BHG-1, (**d**) BHG-2, (**e**) BHG-3, (**f**) BHG-4, (**g**) BHC-1, (**h**) BHC-2, (**i**) BHC-3, and (**j**) BHC-4.

**Figure 15 polymers-14-01017-f015:**
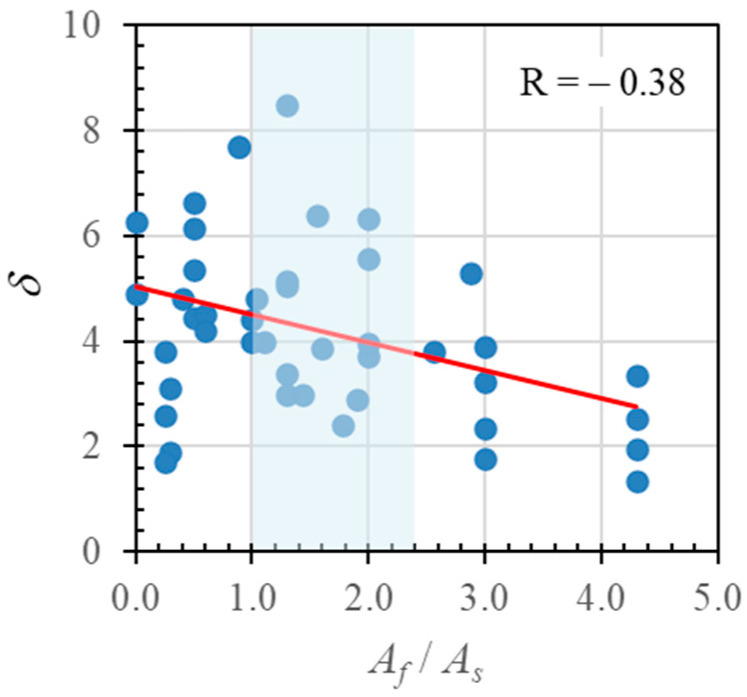
Relationship between hybrid reinforcement ratio (*A_f_*/*A_s_*) and ductility (*δ*) for experimental results, parametric study results, and data obtained from the literature.

**Figure 16 polymers-14-01017-f016:**
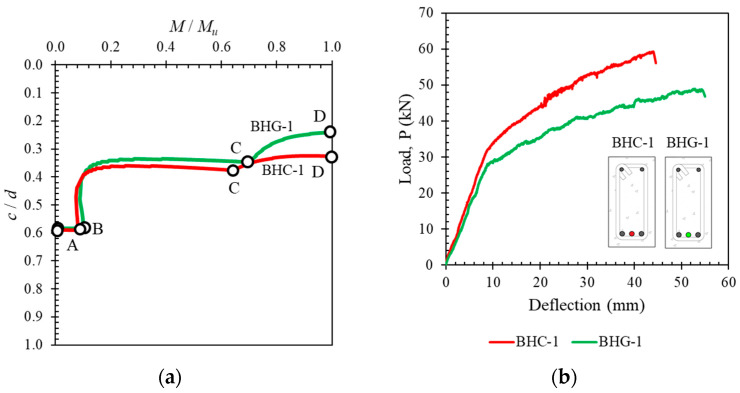
Effect of FRP type shown via (**a**) neutral axis curves and (**b**) load–deflection curves.

**Figure 17 polymers-14-01017-f017:**
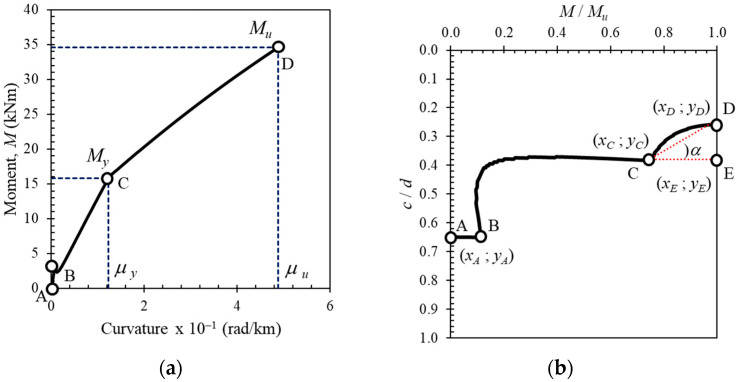
(**a**) The moment–curvature curve obtained from cross-section analysis and (**b**) the corresponding neutral axis curve showing the points used for the calculation of the neutral axis angle (*α*) and displacement index (*δ_N_*) parameters.

**Figure 18 polymers-14-01017-f018:**
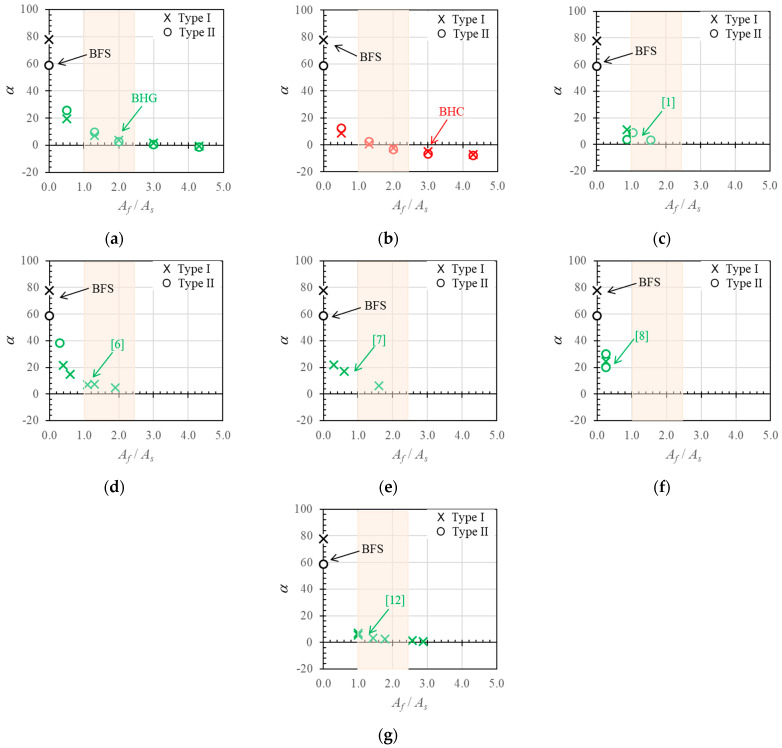
Neutral axis angle (*α*) versus hybrid reinforcement ratio (*A_f_*/*A_s_*), (**a**,**b**) experimental and parametric study, and (**c**–**g**) data obtained from the literature [1,6,7,8,12].

**Figure 19 polymers-14-01017-f019:**
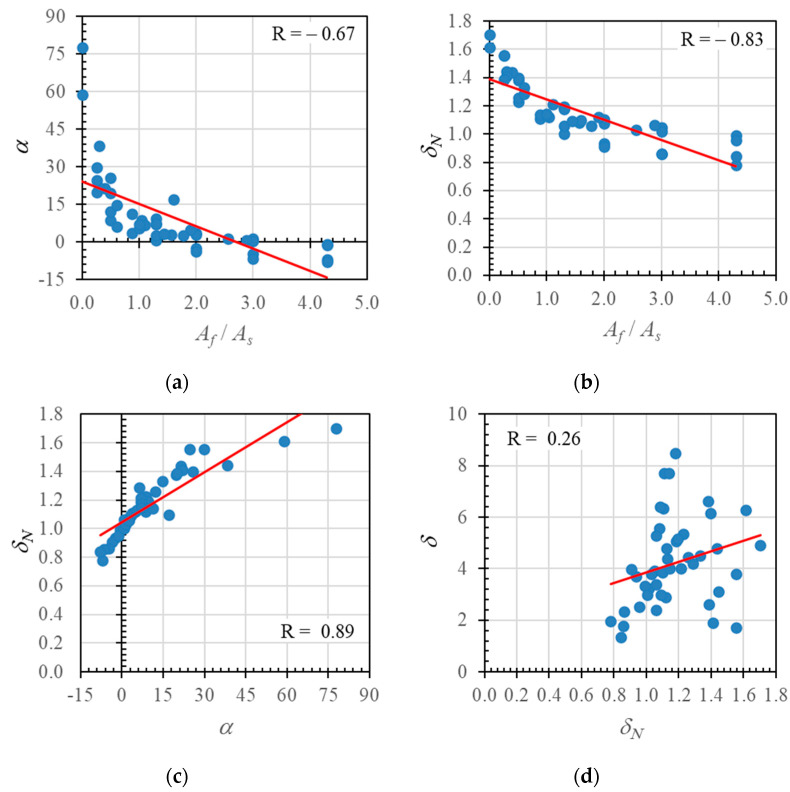
Relationships between (**a**) hybrid reinforcement ratio and neutral axis angle, (**b**) hybrid reinforcement ratio and displacement index value, (**c**) neutral axis angle and displacement index value, and (**d**) displacement index value and ductility, for experimental results, parametric study results, and data obtained from the literature.

**Figure 20 polymers-14-01017-f020:**
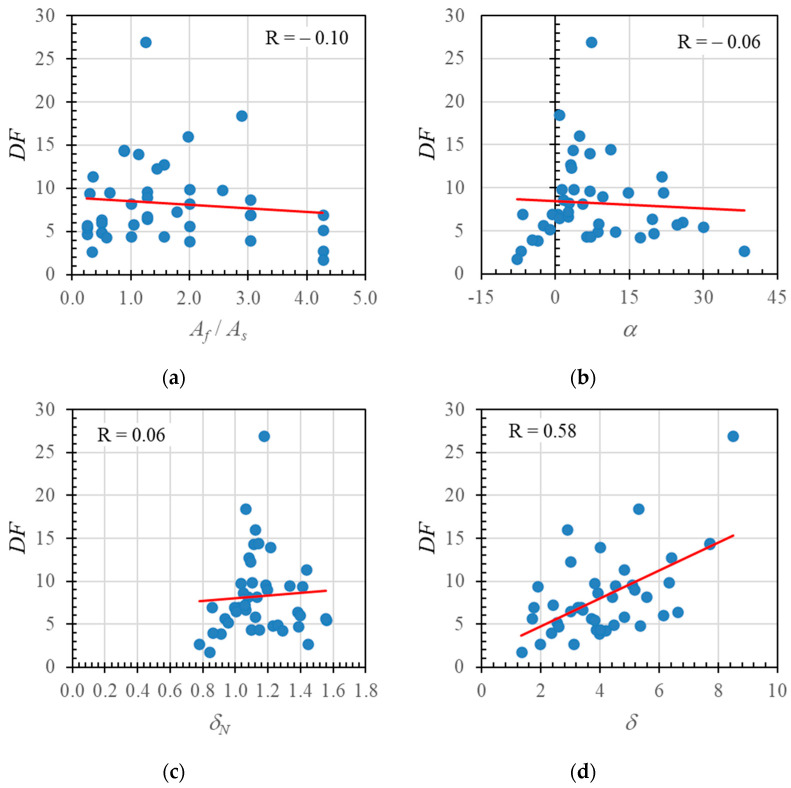
Relationships between (**a**) hybrid reinforcement ratio and deformation factor, (**b**) neutral axis angle and deformation factor, (**c**) neutral axis deformation index value and deformation factor, and (**d**) ductility and deformation factor for experimental results, parametric study results, and data obtained from the literature.

**Figure 21 polymers-14-01017-f021:**
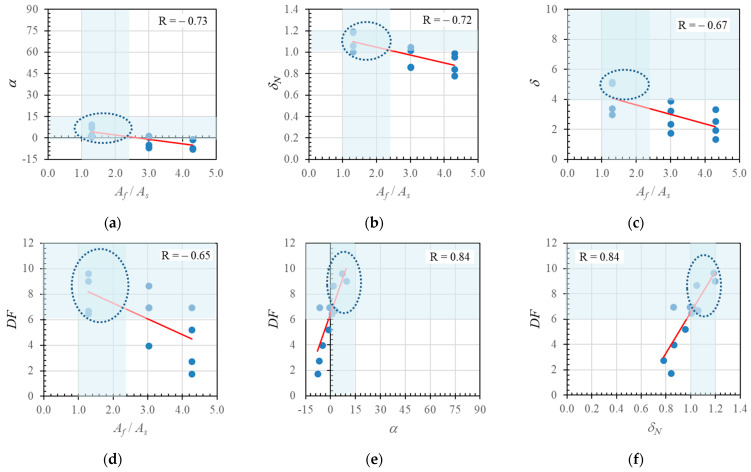
Relationships between (**a**) *α* vs. *A_f_*/*A_s_*, (**b**) *δ_N_* vs. *A_f_*/*A_s_*, (**c**) *δ* vs. *A_f_*/*A_s_*, (**d**) *DF* vs. *A_f_*/*A_s_*, (**e**) *DF* vs. *α*, and (**f**) *DF* vs. *δ_N_* for cross-sections with hybrid reinforcement obtained from the parametric study.

**Table 1 polymers-14-01017-t001:** Material properties of the tested beams and the beams used for parametric study, and corresponding data adopted from the literature.

Beam Notation	Width	Overall Depth	Clear Span Length	*f_c_’*	*f_fu_*	*f_y_*	*d_f_*	*d_s_*	*E_f_*
	(mm)	(mm)	(mm)	(MPa)	(MPa)	(MPa)	(mm)	(mm)	(GPa)
Data from this study [Experimental]									
BFS-2	125	250	2000	20	-	375	-	13	-
BFS-4	125	250	2000	20	-	375	-	13	-
BFG-1	125	250	2000	20	788	-	13	-	43.9
BFG-2	125	250	2000	20	788	-	13	-	43.9
BFC-1	125	250	2000	20	2070	-	13	-	124
BFC-2	125	250	2000	20	2070	-	13	-	124
BHG-1	125	250	2000	20	788	375	13	13	43.9
BHG-2	125	250	2000	20	788	375	13	13	43.9
BHG-3	125	250	2000	20	788	375	13	13	43.9
BHG-4	125	250	2000	20	788	375	13	13	43.9
BHC-1	125	250	2000	20	2070	375	13	13	124
BHC-2	125	250	2000	20	2070	375	13	13	124
BHC-3	125	250	2000	20	2070	375	13	13	124
BHC-4	125	250	2000	20	2070	375	13	13	124
Data from this study [Parametric]									
BHG-5	125	250	2000	20	788	375	16	13	43.9
BHG-6	125	250	2000	20	788	375	16	13	43.9
BHG-7	125	250	2000	20	788	375	19	13	43.9
BHG-8	125	250	2000	20	788	375	19	13	43.9
BHG-9	125	250	2000	20	788	375	16	10	43.9
BHG-10	125	250	2000	20	788	375	16	10	43.9
BHC-5	125	250	2000	20	2070	375	16	13	124
BHC-6	125	250	2000	20	2070	375	16	13	124
BHC-7	125	250	2000	20	2070	375	19	13	124
BHC-8	125	250	2000	20	2070	375	19	13	124
BHC-9	125	250	2000	20	2070	375	16	10	124
BHC-10	125	250	2000	20	2070	375	16	10	124
Aiello et al. [1]									
A1	150	200	2700	45.7	1674	465	7.5	8	49
A2	150	200	2700	45.7	1366	465	10	8	50.1
A3	150	200	2700	45.7	1366	465	10	12	50.1
C1	150	200	2700	45.7	1674	465	7.5	8	49
Qu et al. [3]									
B3	180	250	1800	33.10	782	363	12.7	12	45
B4	180	250	1800	33.10	755	336	15.9	16	41
B5	180	250	1800	34.40	778	336	9.5	16	37.7
B6	180	250	1800	34.40	782	336	12.7	16	45
B7	180	250	1800	40.65	778	363	9.5	12	37.7
B8	180	250	1800	40.65	755	336	15.9	16	41
Lau & Pam [4]									
G0.3-MD1.0-A90	280	380	4200	41.3	588	336	19	25	39.5
G1.0-T0.7-A90	280	380	4200	39.8	582	597	25	20	38.0
G0.6-T1.0-A90	280	380	4200	44.6	588	550	19	25	39.5
Yinghao & Yong [5]									
S2	150	250	1800	80.1	1301	374.5	12	24	75.98
S3	150	250	1800	80.1	1301	374.5	12	24	75.98
S4	150	250	1800	80.1	1301	374.5	12	24	75.98
Refai et al. [8]									
2G12-1S10	230	300	3700	40	1000	520	12	10	50
2G12-2S10	230	300	3700	40	1000	520	12	10	50
2G12-2S12	230	300	3700	40	1000	520	12	12	50
2G16-2S10	230	300	3700	40	1000	520	16	10	50
2G16-2S12	230	300	3700	40	1000	520	16	12	50
2G16-2S16	230	300	3700	40	1000	520	16	16	50

**Table 2 polymers-14-01017-t002:** Failure modes of the tested beams.

Beam Notation	*A_f_*/*A_s_*	First Crack Load	Stiffness	Type of Reinforcement	Reinforcement Ratio	Failure Mode
		(kN)	(kN/mm)			
BFS-2	-	3.6	3.91	Steel	Under Reinforced	SY, CC
BFS-4	-	5.3	3.71	Steel	Under Reinforced	SY, CC
BFG-1	-	4.6	1.33	GFRP	Over Reinforced	CC
BFG-2	-	6.2	1.27	GFRP	Over Reinforced	CC
BFC-1	-	3.7	2.24	CFRP	Over Reinforced	CC
BFC-2	-	3.9	2.21	CFRP	Over Reinforced	CC
BHG-1	0.5	5.5	3.16	Steel and GFRP	Over Reinforced	SY, CC
BHG-2	2.0	3.5	2.23	Steel and GFRP	Over Reinforced	SY, CC
BHG-3	0.5	5.1	2.79	Steel and GFRP	Over Reinforced	SY, CC
BHG-4	2.0	3.6	1.89	Steel and GFRP	Over Reinforced	SY, CC
BHC-1	0.5	5.3	3.71	Steel and CFRP	Over Reinforced	SY, CC
BHC-2	2.0	3.7	2.45	Steel and CFRP	Over Reinforced	SY, CC
BHC-3	0.5	3.2	3.17	Steel and CFRP	Over Reinforced	SY, CC
BHC-4	2.0	6.3	2.39	Steel and CFRP	Over Reinforced	SY, CC

Note: SY = Steel Yielding. CC = Concrete Crushing.

**Table 3 polymers-14-01017-t003:** Ductility (*δ*), neutral axis angle (*α*), yield moment, ultimate moment, and deformability factor (*DF*) of cross-sections with hybrid reinforcement.

Beam Notation	*A_f_*/*A_s_*	*δ*	*α*	*δ* * _N_ *	*M_y_*	*μ_y_*	*M_u_*	*μ_u_*	*DF*	Type of FRP	Type of Cross-Section
					(kNm)		(kNm)				
Data from this study [Experimental]											
BFS-2	−	4.9	77.7	1.7	28.2	1.4	29.1	7.2	5.4	-	Type I
BFS-4	−	6.3	58.8	1.6	25.1	1.4	27.4	6.0	4.8	-	Type II
BHG-1	0.5	6.6	19.6	1.4	21.2	1.3	29.9	5.8	6.4	GFRP	Type I
BHG-2	2.0	6.3	3.6	1.1	14.2	1.2	30.4	5.7	9.9	GFRP	Type I
BHG-3	0.5	6.1	25.7	1.4	20.6	1.3	27.2	6.1	6.0	GFRP	Type II
BHG-4	2.0	5.6	2.8	1.1	13.6	1.5	28.9	5.7	8.2	GFRP	Type II
BHC-1	0.5	5.4	8.5	1.2	24.4	1.4	38.1	4.3	4.9	CFRP	Type I
BHC-2	2.0	3.7	−2.4	0.9	21.2	1.3	41.4	3.9	5.7	CFRP	Type I
BHC-3	0.5	4.5	12.2	1.3	22.7	1.4	33.4	4.6	4.9	CFRP	Type II
BHC-4	2.0	4.0	−3.6	0.9	23.1	1.7	39.9	3.9	3.9	CFRP	Type II
Data from this study [Parametric]											
BHG-5	3.0	3.9	1.5	1.0	15.9	1.2	34.6	4.9	8.7	GFRP	Type I
BHG-6	3.0	3.2	0.5	1.0	15.7	1.5	33.1	4.9	6.9	GFRP	Type II
BHG-7	4.3	3.3	−0.8	1.0	18.4	1.3	38.3	4.3	7.0	GFRP	Type I
BHG-8	4.3	2.5	−1.2	1.0	19.2	1.6	36.8	4.3	5.2	GFRP	Type II
BHG-9	1.3	5.1	7.0	1.2	14.7	1.2	28.7	6.1	9.6	GFRP	Type I
BHG-10	1.3	5.2	9.4	1.2	13.9	1.3	24.6	6.5	9.0	GFRP	Type II
BHC-5	3.0	2.3	−4.8	0.9	26.8	1.4	45.9	3.3	4.0	CFRP	Type I
BHC-6	3.0	1.8	−6.6	0.9	15.7	1.5	33.1	4.9	6.9	CFRP	Type II
BHC-7	4.3	2.0	−7.1	0.8	33.6	1.6	49.6	2.9	2.7	CFRP	Type I
BHC-8	4.3	1.3	−7.8	0.8	38.8	2.1	48.1	2.9	1.7	CFRP	Type II
BHC-9	1.3	3.0	0.7	1.0	19.7	1.3	38.9	4.2	6.5	CFRP	Type I
BHC-10	1.3	3.4	2.6	1.1	17.3	1.3	32.6	4.6	6.7	CFRP	Type II
Aiello et al. [1]											
A1	0.9	7.7	3.6	1.1	8.8	1.9	20.2	11.9	14.4	AFRP	Type II
A2	1.6	6.4	3.1	1.1	10.4	1.9	25.8	10.0	12.7	AFRP	Type II
A3	1.0	4.8	8.7	1.1	20.0	2.2	34.1	7.6	5.8	AFRP	Type II
C1	0.9	7.7	11.2	1.1	9.7	1.8	21.2	11.9	14.5	AFRP	Type I
Qu et al. [3]											
B3	1.1	4.0	6.9	1.2	20.7	1.2	46.0	7.5	14.0	GFRP	Type I
B4	2.0	2.9	4.9	1.1	19.5	1.1	49.9	6.7	16.0	GFRP	Type I
B5	0.4	4.8	21.5	1.4	28.4	1.1	43.4	8.4	11.4	GFRP	Type I
B6	0.6	4.5	14.7	1.3	30.4	1.2	51.9	6.6	9.5	GFRP	Type I
B7	1.3	8.5	7.3	1.2	10.7	1.1	30.8	10.5	27.0	GFRP	Type I
B8	0.3	3.1	38.3	1.4	77.6	1.6	87.3	3.7	2.7	GFRP	Type II
Lau & Pam [4]											
G0.3-MD1.0-A90	0.3	1.9	21.9	1.4	114.9	0.8	166.3	5.0	9.4	GFRP	Type I
G1.0-T0.7-A90	1.6	3.9	6.3	1.1	160.7	1.2	237.3	3.6	4.4	GFRP	Type I
G0.6-T1.0-A90	0.6	4.2	17.1	1.3	195.4	1.1	250.1	3.8	4.3	GFRP	Type I
Yinghao & Yong [5]											
S2	0.3	2.6	19.9	1.4	66.7	1.6	93.1	5.5	4.7	GFRP	Type II
S3	0.3	3.8	29.9	1.6	72.8	1.4	95.2	5.8	5.5	GFRP	Type II
S4	0.3	1.7	24.6	1.6	75.0	1.3	103.2	5.5	5.7	GFRP	Type I
Refai et al. [8]											
2G12-1S10	2.9	5.3	0.7	1.1	15.2	1.4	47.3	8.2	18.5	GFRP	Type I
2G12-2S10	1.4	3.0	3.2	1.1	25.9	1.3	58.4	7.1	12.3	GFRP	Type I
2G12-2S12	1.0	4.4	5.5	1.1	31.1	1.5	55.7	6.8	8.2	GFRP	Type I
2G16-2S10	2.6	3.8	1.3	1.0	31.0	1.4	71.4	5.8	9.8	GFRP	Type I
2G16-2S12	1.8	2.4	2.6	1.1	37.4	1.4	70.9	5.5	7.3	GFRP	Type I
2G16-2S16	1.0	4.0	7.1	1.1	56.5	1.6	81.4	4.7	4.4	GFRP	Type I

**Table 4 polymers-14-01017-t004:** Recommended parameter values of *α*, *δ_N_*, *δ*, and *DF* for cross-sections with hybrid reinforcement.

Parameter	Recommended Value
*α*	>0°
*δ* * _N_ *	>1
*δ*	>4
*DF*	>6

## Data Availability

The data presented in this study are available on request from the corresponding author.

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
