# Peer review of "Ductility Estimation for Flexural Concrete Beams Longitudinally Reinforced with Hybrid FRP–Steel Bars"

_polymers, 2022, doi:10.3390/polym14051017_

Round 1
Reviewer 1 Report
The authors present an experimental study to evaluate the ductility of reinforced concrete beams with hybrid reinforcement, where specifically the effect of reinforcement ratio and the arrangement of the hybrid reinforcement were studied. The position of the bars, ratio of reinforcement and the reinforcement material were varied. The ductility of the beams was evaluated by means of the load-deflection curves from the bending tests. Numerical analysis was then carried out to examine the change in the neutral axis position. All in all, this article is interesting, well structured, the motivation is clear, and the research has merit. The reviewer has a few comments and questions that need to be addressed.
- The abbreviations should be defined in the beginning (FRP, GFRP, BFRP, CFRP and so on).
- “hybrid FRP – steel reinforcement” could also be defined. To the reviewer’s understanding, it is that there is a mix of FRP and steel reinforcement bars in the same beam. This is not necessarily obvious from the name.
- Figure 1 is very pedagogical, with different colors for the different materials of the reinforcement. For even better readability, it could be an idea to write the names of each specimen (BFS-2, BFS-4 and so on) directly below each figure. It is then easier to go back when reading the rest of the papers/graphs.
- It is a bit unfortunate that Table 1 & 3 are spread over 3 pages. Is it possible to divide them in 2 parts?
- Figure 5 could be even easier to read if the cross section with the reinforcement is shown beside the beams.
- Section 4.5 / Figures 10-11. The authors are advised to define c, d, M and
- The wording in conclusion #2 could be rewritten somewhat to be even more to the point. Here also the word “two-zone” is a bit strange.
Author Response
Responses to comments from reviewers can be seen in the .pdf file.

Reviewer 2 Report
The submitted review article “polymers-1567577-v1” entitled “Ductility Estimation of Concrete Beams Longitudinally Reinforced with Hybrid FRP-Steel Bars” is an experimental contribution on the study of Reinforced Concrete (RC) beams with conventional longitudinal tension steel and Fiber Reinforced Polymer (FRP) reinforcing bars subjected to monotonic flexural loading using a four-point bending testing scheme. Paper presents the results of an extensive and well-planned experimental program that consists of 14 RC beam specimens with various reinforcement configurations concerning the position, the ratio and the type of the reinforcing bars. A rather simple and briefly presented analytical study is also included. Paper falls within the scope of the Journal and deals with a topic still open to question. The manuscript is also well-structured and the experimental project is very interesting. Interpretation of the derived results concludes to new and useful findings. As an overall comment and based on the above remarks the paper is worthy of publication after revision. There are some issues, questions and clarifications that should be amended prior publication. The following comments and suggestions are raised for authors’ reference:
- The term “flexural” is suggested to be included in the title as: “Ductility Estimation of Flexural Concrete Beams Longitudinally Reinforced with Hybrid FRP-Steel Bars”.
- In introduction, the literature review is fairly informative. More literature is needed in this part in order to state what has been done and what has not been done in the area of FRP and hybrid FRP/conventional steel reinforced concrete beams. The Authors should focus on more recent and relevant publications. There are many published papers in the same area not provided. The gap between the previous studies and this study should be mentioned. For example, there are some recent and quite relative experimental studies of reinforced concrete beams with carbon-fiber-reinforced polymer bars (in MDPI journals) that could be considered since they include useful comparisons of experimental results for carbon-FRP beams with beams reinforced with glass-FRP bars and comparisons of the experimental results with the predictions according to ACI 440.1R-15 and to CSA S806-12. Further, analytical methodologies for the calculation of the flexural and the shear capacity of concrete members with FRP bars as tensional reinforcement have also been proposed in the following recent studies (order by date) that are recommended to be considered in order to enhance the novelty of the analysis of this research work:
- “Analysis of rectangular hybrid steel steel-GFRP reinforced concrete beam columns”, 2015.
- “Numerical and analytical modeling of concrete beams with steel, FRP and hybrid FRP-steel reinforcements”, 2016.
- “Flexural/shear strength of RC beams with longitudinal FRP bars - An analytical approach”, 2018.
- Although the specific tasks and the objectives of this study are more or less clearly defined, it is recommended to highlight research significance and the subsequent impact of the study on the state of the practice. A systematic literature review based on the aspects of the previous comment would help in this direction.
- What was the control mode used for the beam tests? Load control or displacement control. Further, tests include three displacement measuring points to measure the deflection of the beam. The deflections of all three LVDTs should be presented in the load versus deflection diagrams, at least in some typical tested beams, for comparison reasons. Justification of their locations in the mid-span of the beams is also required.
- The tested beams were a fraction of the real-life size of beams. Can the authors include a discussion on the eminent 'scale-effects' known to influence shear behavior in beams?
- It is suggested to add a new Table to summarize and compare the main test results of all tested beams.
- Equation (1) includes the variable μ which represents the “assuming curvature”. Some details about the estimation of this variable according to the tests would be very useful. Calculations of the strains (ε) of the longitudinal tension steel and FRP reinforcing bars based on the experimental measurements should also be added. According to the this, please provide some explanations how the strain distribution in cross section diagrams in Figure 8 is obtained.
- Proper justification of the adopted stress versus strain of the materials shown in Fig. 4 should be included. Why the neglectable tension strength of concrete is considered in the flexural analysis (Fig. 4a)? Why strain hardening of steel bars has not been considered (Fig. 4b)? How the ultimate strain at failure of the FRP bar has been evaluated and why both CFRP and GFRP bars have the same stress-strain model (Fig. 4c)?
Author Response

(The authors gave the same response as above.)

Reviewer 3 Report
The manuscript by these Authors is an interesting piece about concrete reinforcement by using fiber reinforced polymer – steel. The experimental section comprises a lot of data and the obtained results are worthy of publication, a pity for the authors that are not related to polymers (both like materials and the journal). Through the text, they never used the word “polymer”, they never explain the acronym meaning of FRP, nor other acronyms present in the manuscript. I cannot say neither polymers are in the background in the study, polymers are not present. Thus in my opinion the manuscript is not suitable for publication on Polymers. Before submitting in another journal, I would suggest to the Authors to look at the suggestions reported in the attached .pdf.

Author Response

(The authors gave the same response as above.)

Round 2
Reviewer 2 Report
The revised manuscript “polymers-1567577-v2” with the new title: “Ductility Estimation of Flexural Concrete Beams Longitudinally Reinforced with Hybrid FRP-Steel Bars” has extensively been improved and enriched. The efforts performed by the Authors to consider all the recommendations and to respond to all the criticisms of the previous reviews are greatly appreciated. Thus, the paper is suggested to be accepted for publication in the journal without further re-review from the reviewer. However, some minor improvements are still required, as listed below, that are based on the comments of the previous review round:
Comment 1. Well-addressed, hence, further revision is not required.
Comment 2. Well-addressed, hence, further revision is not required.
Comment 3. Well-addressed, hence, further revision is not required.
Comment 4. Well-addressed, hence, further revision is not required.
Comment 5. Although the influence of the “scale-effects” on the shear behavior of reinforced concrete beams have not been studied in this study, and the authors do not have data to discuss this topic in the text, it is suggested to add a few findings from recent articles of the literature that examine this issue. Papers concerning the evaluation of the flexure/shear design equations of FRP-reinforced concrete beams with or without shear reinforcement wouls help in this direction. Nevertheless, some concluding remarks concerning this issue would promote the paper.
Comment 6. Table 2 has been enriched. However, this Table still seems incomplete. It is suggested to add more numerical test results concerning the values of strength at first cracking and at ultimate, corresponding displacements, stiffness, ductility, etc. This would be very helpful for the readers of this interesting article.
Comment 7. The Authors responded that “Figure 8 is purely obtained from a numerical cross-sectional analysis with the help of a computer program”. According to this statement, some explanations are required about the formulation and calculation of the strain distribution in cross section.
Comment 8. Well-addressed, hence, further revision is not required.
Author Response
Comments from authors can be seen in the attached file.

Reviewer 3 Report
My concern is related to the suitability of the manuscript topic with the scope of the Journal. The Authors have not at all reiterated this observation, therefore I remain of my opinion: the manuscript is scientifical of a good standard but not suitable for polymers because it does not strictly concern this topic.
Author Response

(The authors gave the same response as above.)
